# Introducing the World’s First Global Producer Price Indices for Beef Cattle and Sheep

**DOI:** 10.3390/ani11082314

**Published:** 2021-08-05

**Authors:** Mohamad Isam Almadani, Peter Weeks, Claus Deblitz

**Affiliations:** 1Thünen Institute of Farm Economics, Bundesallee 63, 38116 Braunschweig, Germany; claus.deblitz@thuenen.de; 2Weeks Consulting Services Pty Ltd., St Ives, Sydney, NSW 2075, Australia; weeksconsultingservices@gmail.com

**Keywords:** producer price index, finished cattle, weaner cattle, lambs, sheep

## Abstract

**Simple Summary:**

Global finished cattle and sheep production and prices are becoming increasingly volatile due to various demand, supply, animal disease, and environmental factors. With beef and sheep production systems being globally connected, an approach monitoring the developments of global meat prices as received by producers becomes increasingly important. This paper presents four global price indices for beef and sheep meat based on prices received by livestock producers obtained from 33 beef and 20 sheep producing countries. These indices are: finished cattle, weaner cattle and lambs producer price indices, as well as a lambs and sheep producer price index—all using an internationally standardized approach. These global indices and associated country and regional indices are used to monitor and explain developments in global beef cattle, lambs, and sheep prices over the index period 2000–2019. The global farmgate cattle and sheep prices have been mainly impacted by developments in global supply and demand, as well as changes in international market access. However, factors which are more local in nature, such as droughts, diseases, local economic developments, exchange rates and policy interventions, have also had a considerable impact on price developments.

**Abstract:**

While international beef and sheep meat price developments are usually measured with meat trade prices (provided by FAO), no comparable information exists on world average of national prices that producers receive for livestock. This paper aims to fill this gap by introducing a set of global producer price indices representing cattle, lambs, and sheep prices as received by producers: the agri benchmark of weaner cattle, finished cattle, lambs and lambs and sheep price indices. These Laspeyres, production-weighted indices measure changes in global farm gate prices as provided annually by the agri benchmark Beef and Sheep Network, with this paper covering prices between 2000 and 2019. The results showed that growing Asian imports, local economic developments in South America, the interconnection with the dairy sector in Europe, growth of beef consumption in China and exchange rates shifts are the key factors that drove developments of global beef producer prices over the past 20 years. Droughts in Oceania and the rapid rise in China’s sheep meat prices are highly reflected in the Global Lambs and Lambs and Sheep Meat Price indices. The indices indicate whether cattle and sheep producers globally are receiving more, or less, for the commodity and may increase or reduce production and investment accordingly. This will be of more use if there were similar producer price indices for competing enterprises, such as dairy and cropping, and for competing proteins, such as pigs, poultry, and fish.

## 1. Introduction

Global meat production has increased rapidly in half a century and has almost quadrupled since the late 1960s, mainly driven by poultry and pork production. Global cattle herd and sheep flock have risen by a relatively slow 14% and 16% over the last two decades, respectively. The largest growth in cattle inventory was observed in Africa (up 55% in 2019 compared to 2000) followed by South America (up 20% over the same period). Both regions present an equal share of the global cattle inventory of 24% each. While Oceania recorded a sharp decline in the sheep flock of 42% since 2000, Asia’s sheep inventory grew by 27% [1].

Global demand for meat is also growing. Population growth, urbanization, and the rise of a global middle class as a result of income growth particularly in low- and middle-income countries, have hastened a dietary transition toward higher consumption of meat products [2]. This demand growth is expected to continue over the coming decade, providing an incentive for the global livestock herd and flock expansion and increased productivity needed to increase supply and consumption. Over the next decade, global meat consumption is projected to increase by 12% compared to the 2017–2019 average [3].

In the last 20 years, the global meat industry has been experiencing complex interactions between various major developments such as widespread severe droughts, animal disease outbreaks, trade policies and agreements, investments, and technological innovations [3,4]. Furthermore, beef and sheep meat production systems are becoming increasingly impacted by a range of sustainability challenges, such as global warming, increased competition for land and water resources, environmental regulations, animal welfare considerations, consumer perceptions for meat products, and industry structure and related policies [5,6,7]. These aspects may give rise to a more volatile sector over coming years regarding cattle and sheep inventories, productivity, production, and prices.

In the last two decades, food price volatility has been above the levels observed before 2000 [8,9,10]. Beef and sheep meat prices, measured as nominal, not inflation-adjusted, prices of food commodities traded in international markets and presented by the FAO Bovine and Ovine meat indices, showed three clear peaks in 2008, 2011, and 2014 and remain well above the low levels of late 1990s and early 2000s [2,9]. Among others, product prices are crucial variables in making decisions pertaining to production. Analysis related to developments in producer prices requires clarifying price trends (price levels) and the volatility around those trends (cycles and extreme events) [8]. In this context, and with beef and sheep production systems being globally connected, an approach monitoring the developments of global meat prices as received by producers becomes increasingly important.

The producer price index (PPI) is a commonly used approach to measure changes in a set of prices of producer goods and services over time. The price index approach helps to aggregate a large number of prices and quantities for several goods or services into a scaler for the purpose of knowing the extent to which these prices have changed [11,12]. It assumes a value of 100 for a selected base period, then values of the index for the observed period of time present the percentage change in prices from the base period. The *Producer Price Index Manual* issued by the International Monetary Fund (IMF) in 2004 [13] highlighted the importance of the Producer Price Index (PPI) as an analytical tool required by researchers, national statistics’ offices, and international organizations, such as Eurostat, the Organization for Economic Co-operation and Development (OECD), the Food and Agriculture Organization of the United Nations (FAO), and European Central Bank (ECB). It is a key economic indicator that can be used for economic analysis and decision making and to measure the risks introduced by price instability. Among others, the PPI can be used to monitor market developments and competition, food security concerns, inflationary trends, and price inflation transmission from the production level to the retail sector [13,14,15,16].

Regarding agricultural products, the FAO Food Price Indices (FFPIs) were introduced in 1996 as an international approach to help monitor the monthly change in the global prices of 23 food commodities traded in international markets. Meat products are represented in the FFPIs in four different indices; bovine, ovine, pig, and poultry [9,17]. As for all FFPIs, the FAO Meat Price Indices are Laspeyres price indices calculated using export prices of the selected meat products as trade-weighted averages based on export values over a chosen three-year base period [15,16,17].

The FAO Bovine Meat Price index is calculated based on export prices of specific beef products commonly exported by the USA, Brazil, and Australia as a weighted average based on export value shares in the 2014–2016 base period. The FAO Ovine Price index represents the price of New Zealand export lambs of 17.5 kg carcass weight and Australian medium trade lambs of 18–20 kg carcass weight [18]. Thus, these indices can be used to monitor developments of global meat prices as export prices of international traded commodities rather than production prices as received by farmers. However, these indices have often been cited as global producer price indices due to the absence of indices representing global producer prices. Weighing the average by export value shares for three producer countries (in the case of beef) may not appropriately capture trends in regional meat production prices around the world. Besides, the actual prices received by producers, particularly in developing countries, may be quite different from the international market or border prices, as their transmission from the farm gate can be influenced by local factors, especially subsidies, market interventions, and trade barriers and policies [8,16,17].

The output PPI measures price changes of outputs produced by businesses over time. Therefore, the evaluated prices should be the sale prices of products as they leave the producer [13]. The agricultural producer price index that adequately reflect prices received by farmers should be calculated based on “prices received by farmers in return for the products they deliver, at a stage as close as possible to the farm’s exit” [19]. Similarly, the IMF in 2004 clearly pointed out that “for many agricultural products the prices collected should be “farm gate” prices—that is, the per unit prices received by the farmer for each product sold as it leaves the farm” [13], p. 79.

Against this background, this study aims to (i) introduce a set of world average producer price indices for cattle, lambs, and sheep based on national prices received by livestock producers at the farm gate obtained from 33 beef and 20 sheep producer countries (named global producer price indices in this article), (ii) harness these indices to understand key factors leading to the developments in global beef and sheep meat prices over the index period (2000–2019) based on national and regional indices, and (iii) highlight the relativity of these producer livestock price indices to the FAO bovine and ovine meat price indices. The study distinguishes two indices for beef finishing systems (finished cattle and weaner cattle) and two indices for sheep farming systems (lambs plus lambs and sheep). Such a detailed set of indices can help to (i) reflect the structure of beef and sheep prices in producing countries which are not within of the main exporters, (ii) represent what the medium and small producer countries produce, and (iii) reflect country-specific drivers of prices such as policy interventions and supply and demand changes in major producing and consuming countries. This in turn can help to understand the key drivers of current and future trends and developments in global beef and sheep producer prices. The hope is that producers, agri business throughout the supply chain, industry associations, governments and global, regional, and country agencies will find these indices useful to observe relative movements in prices over time. These observations can provide producers with management and investment decisions and provide to agri business and industry and government policy making and strategic planning, as movements up or down in relative prices can indicate a shift in relative competitiveness and a need to alter policies, investments, management, or marketing. The indices alone are not sufficient grounds on which to base investments or divestitures, hence our elaboration of the range of underlying causes for changes in relative indices (globally, regionally, and between countries).

## 2. Materials and Methods

### 2.1. The Global PPIs Identified in the Study

This paper introduces four Global PPIs that measure changes in global farm gate prices of final outputs produced by cow-calf breeding, cattle finishing, and sheep farming systems between 2000 and 2019. These indices are production share weighted Laspeyres indices—the Finished Cattle Producer Price Index (FPPI), Weaner Cattle Producer Price Index (WPPI), Lambs Producer Price Index (LPPI) and Sheep Meat (lambs and mutton sheep) Producer Price Index (SPPI).

FPPI, LPPI, and SPPI focus on the final meat products of beef finishing and sheep farming systems as animals that are ready for slaughter and not as intermediate products. For instance, the FPPI does not consider cattle that are produced in a farm or production system (e.g., weaners) and sold to be finished in another one. The indices highlight two different outputs as final products in sheep farming systems—lambs and sheep meat. Several countries provide production and price data regarding lambs being finished and traded as independent products; thus, LPPI can be calculated. SPPI is adopted to cover all countries, including those countries that do not treat lamb finishing separated from the ewe husbandry, but they provide data as sheep meat produced involving both lambs and mutton sheep.

To enable the observation of cattle prices earlier in the cattle production chain, WPPI focuses on cow-calf production systems that mainly produce weaner calves as an intermediate product into the finished cattle production chain. WPPI measures price changes in weaner cattle sold to beef finishing systems. Thus, WPPI does not reflect prices of weaners which are sold off for meat production at the age of weaning. The study indexes WPPI separately because in most cattle rearing production systems cow-calf production is a self-standing business principally aimed at producing weaners. The cow-calf enterprise starts with the birth of the calf and ends with the day of weaning. Then, the beef finishing business starts when weaner calves (or dairy calves and feeder cattle) are bought from outside the farm or transferred from the cow-calf (or dairy) enterprise to the beef finishing enterprise on the same farm [20]. Within the entire cattle finishing business, WPPI can be classified as an intermediate input price index [13]. However, since prices used to calculate the WPPI are weaner prices, as the final output from cow-calf farms, then WPPI can also be described as an output producer price index.

### 2.2. Countries Involved in the PPIs and Data Sources

PPIs are calculated based on the global agri benchmark Beef and Sheep Network Data. The agri benchmark Beef and Sheep Network was established in 2002 as a global, non-profit network of producers and agricultural experts from 50 institutional partners representing finished cattle and sheep value chains across the world.

FPPI and SPPI are represented by 33 and 20 countries, respectively. The knowledge of finished cattle and sheep production systems and the access to producers are the main criteria for selecting the network partner institutions and individuals [20]. Since only certain countries treat lamb finishing as an independent business, LPPI is represented by 11 countries. WPPI is represented by 20 countries; thus, it does not cover all 33 countries that are included in the FPPI. The reason is that the weaner production business is not as relevant in dairy specialized countries where dairy calves are the prevailing cattle for finishing, not beef-bred weaner calves. To our best knowledge, this approach has the largest geographic coverage of any regional or international index. Table A1 (in Appendix A) provides a list of the countries included in each index.

The agri benchmark Network Partners, who are members of the agri benchmark partners’ institutions, provide sector and farm data on cow-calf, finished cattle, and sheep farming systems in their countries on an annual basis (calendar year) based on national statistics. At the sector level, agri benchmark partners provide annually updated data regarding national livestock inventories (total cattle, sucker-cows, cattle on feed, and sheep and lambs), quantities of total produced meat (beef, lamb, and mutton), export and import volumes, and values and unit price of each product. This data set is referred to as the agri benchmark database in this article [21]. Annual unit production prices and the production volume, for weighing the index, are needed to calculate the PPIs. The national production of beef, lamb, and sheep meat are estimated as carcass weight equivalent. It is meaningful to reflect the dressing percentage in each country by using the carcass weight equivalent since producers sell their finished animals that are ready for slaughter; thus, the real evaluated product is meat. However, the weaner cattle production is estimated as live weight equivalent because it references live cattle that still have a further finishing period before being slaughtered.

The PPIs are a world average of the price of cattle, sheep, and lambs (on a USD/kg carcass weight basis) that are typically produced in each country. Hence, for accurate and consistent PPIs, representative price sampling from a set of well-defined products is required. This can be fulfilled by identifying the most prevailing products produced from regional hot spots in terms of production in each country. The network partners in each country facilitate the selection of the most prevailing product type for beef finishing and sheep meat that are nationally classified as typical production sold at farm gate in their countries. The main criteria to identify regional hot spots for livestock production are the total animal numbers per region and animal numbers per hectare of agricultural land (density) [20]. In cases where there is more than one regional hotspot or various prevailing products in a country, the price average of these products and regions is taken. For example, prices of sheep meat in Brazil are calculated as an average of ewe northeast and ewe south prices. Table A2 (in Appendix A) highlights product types whose prices have been chosen as being most representative of beef and lamb production in each index country.

This paper and project specifically aim to develop price indices that reflect an average of typical animals across the world, as that is what is relevant to each country and globally. This is not to dismiss the fact that there are significant differences between what is typical (as listed in Appendix A Table A2). These variations arise from differences in production systems, climates and environments, resources, breeds, industry concentration, and structure and asset fixity across different localities country. As a result, specific product characteristics and animal and meat “quality” varies too, although what is seen as “high quality” animals and meat in one country can be different from another.

Prices involved in the FPPI, LPPI, and SPPI refer to the price received by farmers for the whole carcass produced and sold without distinction between cuts. The prices used are the amount receivable by the producer from selling a unit of output; hence, subsidies on products are excluded from producer returns of the product. In terms of transport costs, our approach follows the principle presented in the revised System of National Accounts in 1993 (published jointly by the United Nations, the Commission of the European Communities, the International Monetary Fund, the Organization for Economic Cooperation and Development, and the World Bank) indicating that transport costs will be included in the product price as long as they are included as part of normal business practice and are not separately billed by the farmer to the recipient [22], Chapter III, p. 95.

Prices are expressed in United States dollars (USD) at market exchange rates and analyzed as nominal prices, not inflation-adjusted prices. Having the global PPIs calculated in USD will facilitate the comparability between the global PPIs and the other commodity at national, regional, and global levels. Furthermore, the USD is a logical choice given that the USA is one of the largest beef producers, consumers, importers, and exporters of beef and the USD is the currency forming the basis of the FAO international meat price indices.

The global PPIs are indicators of whether producers globally are receiving more or less for the commodity (cattle, lambs, and sheep) and may increase or reduce production and investment accordingly. Thus, the indices do not aim to evaluate the absolute production prices, rather they reflect the extent to which these prices have changed over the index period. This target can be captured as long as the local and the USD price series are significantly correlated in the producer countries. The exchange rate correlations, USD prices, and local currency prices for each of the key producer countries with high production shares will be explored in the results and discussion (Section 3).

Production and prices data are updated annually in the month of February jointly with research partners in the participating countries.

### 2.3. Price Index Formula

The global PPIs introduced in this study are output indices calculated on the basis of the Laspeyres approach. The Laspeyres index was first proposed in 1871 [23] and is still widely used today for both producer and consumer price indices [13,16,19,24,25]. It is commonly used by statistical offices in the national and international organizations worldwide, such as the FAO and the National Institute of Statistics and Economic Studies in France (INSEE). All OECD Member countries use the Laspeyres approach for their published producer price indices [26]. The FAO seafood trade statistics used the Laspeyres formula to introduce a global fish price index for the first time in 2012 [27].

Given *n* products in the basket, the Laspeyres approach denotes the percentage changes in the product prices sold in an observed time (month or year) and weighting them by the total production value in the base period (Formula (1)). Statistical agencies have traditionally used the Laspeyres index as their target index under the ease of implementation. It facilitates timely updates since the weights to be applied (commonly revenue data) are required only for the base period [13,17]. This enables an early index update for monitoring and assessing the most recent agricultural market developments at the global level [14]. This is more convenient than indices’ approaches that use weights from the current period, such as the Paasche index.

Laspeyres Formula (1). Production value weights.
(1)$$P_{L}=\overset{n}{\underset{i=1}{\sum }}(P_{i}^{t}/P_{i}^{0})s_{i}^{0},$$
$P_{i}^{t}$: the price of the *i*th product in year *t*$P_{i}^{0}$: the price of the *i*th product in the base year.$s_{i}^{0}$: the production value share of the *i*th product in the total production value of all products in the basket in the base period, that is: $s_{i}^{0}=P_{i}^{0}q_{i}^{0}$/$\sum_{i=1}^{n}P_{i}^{0}q_{i}^{0}$.$q_{i}^{0}$: the production quantity of the *i*th product in the base year.


The Laspeyres index compares the revenue of a basket of products in the base period with the revenue in a given period of the same basket of products. Since the global PPIs introduced in this paper evaluate one product by each index, the basket of products refers to the same product type (beef, lamb, sheep meat, or weaners) but from a group of countries. Moreover, the global PPIs cover many countries which greatly differ in terms of price levels. Therefore, weighing the indices by the countries’ share of the production value (production quantity × price per unit) involve disadvantages: (i) double-counting the price—with the result that high-priced countries, such as China, will also have a high value (weighting) and vice versa; (ii) the introduction of a source of error as any over- or under-estimation of the average country price is compounded by it being used both in the price and the weight applied to it; and (iii) product prices (and hence production value) are more volatile than production volume, making it hard to set a base period that is representative for all four PPIs and for all major producing countries. The PPIs base period 2014–2016 (which will be described subsequently in more detail), saw peaks in cattle prices in many countries (including the USA and China), and hence value, whereas production was more representative and stable.

For these reasons, the global PPIs are calculated based on the production-weighted average from all countries based on production quantity shares in the base period (Formula (2)).

Laspeyres Formula (2). Production quantity weights.
(2)$$P_{L}=\overset{n}{\underset{i=1}{\sum }}(P_{i}^{t}/P_{i}^{0})s_{i}^{0},$$
$P_{i}^{t}$: the production price in the *i*th country in year *t*.$P_{i}^{0}$: the production price in the *i*th country as an average price over the three-year base period 2014–2016.$s_{i}^{0}$: the share of the production volume of the *i*th country in the total production volume of all countries in the base period, that is: $s_{i}^{0}=q_{i}^{0}$/$\sum_{i=1}^{n}q_{i}^{0}$.$q_{i}^{0}$: the production quantity in the *i*th country as an average over the three-year base period 2014–2016.


The indices monitor the annual price developments over the period 2000–2019 as a production-weighted average from all countries based on production quantity over a chosen three-year base period 2014–2016. A three-year base period is chosen to minimize the impact of variation in product quantities. Producer or consumer price indices typically use a single year as a base. However, weighing the index based on several years’ data can greatly reduce the impact of any seasonal or abnormal variance in weights due to economic or environmental conditions that often impact product quantities and prices [13,17].

A three-year base period 2014–2016 is adopted for all indices in this study because it can be considered the most representative period for most countries in the past two decades, as: (i) it is recent and yet production data is unlikely to be subject to significant revisions, (ii) the production of the main producing countries are relatively stable relative to their trend volumes, thus are structurally representative of recent years, and (iii) it matches with the FAO meat price indices base period to allow easy comparison between our global producer livestock price indices and the FAO export meat price indices.

## 3. Results and Discussion

### 3.1. The Global Finished Cattle Producer Price Index FPPI

The Global FPPI includes weighted finished cattle prices from 33 countries, with all the major producing countries included except India, representing approximately 76% of global beef production in 2019 [1].

As shown in Figure 1, the Global FPPI rose appreciably in all bar one year from 2003 to 2011. It stabilized over the 2011–2013 period before reaching its peak in 2014 at an index value of 107, 152% above the starting period of 2000–2002. This rise in the FPPI primarily reflects the rapid growth in demand for beef in Asia and associated beef import liberalization (especially in China, Japan, and Korea), together with slowing global beef production growth. Underlying factors driving Asian demand have been population and income growth, urbanization, and the development of meat supply chain infrastructure (especially cold storage and modern retail and foodservice). At the same time, beef supply growth has been constrained by resource and environmental restraints, rising input costs, falling dairy cow inventories (especially in Europe), and slow productivity improvements. By 2016, the index had fallen back by 13% from the 2014 peak. It declined slightly in 2018 and 2019 but was still at a historically high level in 2019—119% above that of the starting period in 2000–2002.

In order to better understand global finished cattle price developments over the observed period, sub-indices for countries and regions that contributed highly to the average global production over 2014–2016 were calculated. Four sub-indices were identified: North America, South America, Europe, and China price indices. These countries and regions accounted for nearly 88% of production weightings in the global FPPI as an average over the base period (Table 1). Global meat production and prices vary between countries due to their differences in resource base, input prices, environmental conditions, industry structure, exchange rate against the USD, global export and import access, and government supports. Nevertheless, presenting a group of countries under a sub-index is warranted given the positive and significant correlation of price changes between countries within each identified region.

#### 3.1.1. The North America FPPI

North America includes the US, Canada, and Mexico and accounts for 28% of production weightings in the Global FPPI as an average over the base period, of which 22% is US production. Therefore, the US is the focus of the following analysis. Cattle producer prices in Canada and Mexico are highly impacted by prices in the US—accentuated by geographic proximity, large trade in cattle and beef under the North American Free Trade Agreement (NAFTA), and the complementarity of their cattle and beef sectors to those of the United States. Canada and Mexico are the only significant cattle suppliers to the US market—destined both for immediate slaughter and for US feedlots for finishing [28].

The North America FPPI moved closely with the Global FPPI over the 2000–2019 period, both increasing to a peak in 2014, before falling back to 2016 and almost stabilizing since (Figure 2). However, the North America FPPI is more volatile than the global FPPI as is to be expected for all country indices when compared with the aggregated global one.

The North America FPPI has also risen much less than the global FPPI over the index period as a whole. This is evident in the 2004 to 2008 period due to the outbreak of bovine spongiform encephalopathy (BSE or mad cow disease) and resultant BSE-related restrictions on US imports into Japan, Korea, and other Asian markets. Those restrictions were eased in 2006, largely removed in 2013, and effectively removed completely in 2019 [29,30]. However, US prices never regained what they lost during this period against the global prices. This is probably due to the expansion in Asian beef demand, prices, and imports (particularly China, but also South East Asia and Korea) that has contributed directly to the rise in the global FPPI (China and Indonesia have among the highest cattle prices in the world) and benefited other beef exporters more than it did in the US (especially South America and Australia).

Over the top of the global trends, as outlined above, is the influence of the US cattle cycle on the North America FPPI. The United States cattle herd has a cyclical nature in response to actual and expected changes in profitability of beef production. The cattle cycle is also highly influenced by climate conditions which, in turn, impact calves born and the time needed for raising calves to market weight. The last full cattle cycle, a 10-year cycle, began in 2004 with 3 years of herd growth followed by a 7-year liquidation period until 2014. The cattle herd contraction started in 2007 due to increasing feed and energy prices and extended up to 2010, when widespread drought began and hit varying regions of the United States until the first half of 2014. Dry conditions forced producers to cull cows and limit heifer retention, which reduced the following year’s supply of calves placed in feedlots. This in turn drove down the size of the national cattle herd. In 2014, the national herd and suckler-cow inventories had fallen by 6% and 11%, respectively, compared to the cycle starting year 2004 [21,28,31,32].

In 2015, the beef industry saw the first increase in cattle and calf production since 2007, announcing a new cattle cycle by herd rebuilding during which females were retained for breeding rather than being slaughtered [33]. The cattle and calf inventory rose by 1% in 2015 compared to a year before, whereas cattle slaughter declined by 5%, keeping producer prices at 2014 levels [21]. The cattle and calf inventory in 2016 was 3% above the 2015 inventory with a high of 32% of heifers retained for beef replacement [21,32]. The total herd continued growing year-on-year and reached 94.8 million head in 2019, 7% above 2014 levels, with 1.35 million more placements compared to 2014 [21]. The 2015–2019 growth was the fastest four-year growth since 1973–1977. This continues to be the primary headwind to higher US producer prices. However, the strong domestic and international demand for US beef has continued to provide price support since 2011, preventing the normal cyclical fall in producer prices.

#### 3.1.2. The South America FPPI

Figure 3 presents the South America FPPI in comparison to the global index. South American countries involved in the FPPI accounted for about 28% of production weightings in the global FPPI as an average over the base period. The beef sector in South America represents the third-largest agricultural industry in the region by gross value of production, behind soybeans and chicken meat [1].

The South America FPPI is dominated by Brazilian beef producer prices due to the high production share of Brazil—accounting for 65% of South American production. USD and domestic finished cattle prices in Brazil are significantly correlated over the index period (ρ value ˂ 0.01).

While trends in the South America FPPI have been similar to those in the global FPPI, it has been more volatile and driven by big local economic and political developments (as the vast majority of production is consumed on local markets), associated large shifts in the value of the local currencies against the USD, market access changes, and changes in local supply.

The South America FPPI rose along with the Global FPPI in the first 12 years of the index period. The five years from 2007 to 2011 saw South American prices rise faster than global prices—due to a combination of strong economic growth resulting in rising domestic demand, associated stronger currencies against the USD, and growth in export demand, principally from Europe and Asia. However, while global prices peaked in 2014, South American prices have been falling since 2011. This reflects the rapid deterioration in economic growth, political stability, and the currencies since then. Annual GDP growth in Brazil peaked at around 7.5% and Argentina at 10%, both in 2010, and the local currencies peaked around the same time (Brazil real in 2011 and Argentine peso in 2008). GDP growth rapidly declined thereafter and has been in, or close to, recession in Brazil since 2014, Argentina since 2012, and Uruguay since 2015 [34,35]. The Brazil real is now over 70% below its peak in 2011, while the Argentine peso is over 50% below the 2011 peak [36] (Figure 4).

Such a rapid deterioration in economic growth and political conditions has severely affected consumer demand for beef while rampant inflation and currency devaluations have lifted beef farm input costs. The fall in currency values has a direct and proportionate impact on the South American FPPI, as this index is calculated in USD.

The currency declines helped to boost producer prices in local currency terms. Value determination and adjustment in the exchange rate as a crucial macro price may lead to a large impact on world export prices of certain products. This can be observed in developing countries by expanding exports through currency devaluation and vice versa [37], Chapter 10. The recent currency declines in Argentina and Brazil were assisted by improving global import demand and improved access to China and the US. China suspended its ban on imports of Brazilian beef in May 2015 [38] and the US government allowed limited access to Brazilian beef in 2016 from FMD-free areas with vaccination in 14 of Brazil’s 27 states [39].

Despite recent investments in on-farm productivity, much of the Brazilian beef industry remains highly vulnerable to any adverse conditions resulting from economic, political, climatic, environmental, and global trade developments. Cattle production in Brazil is mostly grass fed and characterized by small producers in the north and west of the country where technologies are not well developed and utilized. These producers are managing the majority of the Brazilian herd and principally supply the large domestic market. Beef production in these regions is subject to natural resource constraints and environmental conditions and policies.

Drought also has an extreme impact in pasture production systems. Drought across Brazilian pasturelands in 2014 and 2015 increased the number of slaughtered animals in 2015, resulting in higher beef supply and lower prices—by 17% compared to 2014. This was followed by two years of cattle rebuilding, increasing beef prices by 8% in 2017 from 2015 [38,39,40,41].

#### 3.1.3. The Europe FPPI

Thirteen countries are included in the Europe FPPI, representing 20% of the base period production share over all countries involved in the global FPPI. The Europe FPPI is highly impacted by Russia, which has around 30% of production among the European countries, followed by France and Germany with 15% and 11% shares, respectively. USD and domestic prices are significantly correlated in Russia, France, and Germany over the index period; the ρ values are ˂ 0.01 in the case of Russia and ˂ 0.001 in the case of Germany and France. Despite important regional differences in terms of climate, pasture availability, breeds and animal types, and cattle finishing farming practices, the movements in finished cattle prices across European countries were highly and significantly correlated over the past two decades.

As shown in Figure 5, the Europe FPPI showed a much smoother development and lower price variability than the North America and South America indices. Finished cattle prices in Europe showed a steady upward trend from 2002 to 2012 and the index almost doubled during the last 20 years. The Europe FPPI initially rose faster than the global FPPI, as production fell and the euro and Russian ruble values rose against the USD—to reach its highest levels between 2011 and 2014, at 120 index points in 2012. It then fell to around 96 in 2015, following dairy reforms and a devaluing Euro against the USD, and continued around this level for the remainder of the observed period. The sharp fall in the Russian ruble in late 2008 and again in 2014 is reflected in substantial falls in the Europe FPPI in the following years.

The majority of beef produced in Europe is of dairy origin. This is particularly evident in the ratio of dairy cow to suckler cow inventories, which are estimated to have been 60:1 in Russia over the last three years. This ratio is also in favor of dairy cattle in most of the EU countries: 10:1 in Poland, 6:1 in Germany and Italy, 3:1 in Austria, and 1.3:1 in Ireland and the UK. It is slightly reversed in France at 0.9:1, and to a large extent in Spain and Portugal at around 0.4:1 [21,42].

The Russian meat sector has been undergoing a downward trend in cattle inventories. The economic downturn after the collapse of the Soviet Union in 1991 had an enormous impact on beef production in Russia. Farm restructuring, subsidy removal, and input price liberalization caused the total cattle herd to fall by half between 1992 and 2000, from 55 million to 28 million head [1]. Governmental measures that were applied during the 2000s to support livestock producers and develop meat cattle breeding have been unable to prevent the continuing decline in the cattle herd to nearly 18 million head in 2019 [1,43]. The decrease in livestock inventories occurred because Russian dairy farmers continued to offload cattle to slaughter at a faster pace than could be replaced through reproduction. This can be attributed to increased milk cow productivity and low profitability of finishing animals [44]. The recent economic growth in the country supported consumer affluence and meat consumer demand. Beef producer prices in Russia increased by nearly 235% over the observed 20-year period—reflecting the fall in supply in the first decade and lift in demand in the second. Due to weak consumer demand for high quality beef, specialized beef producers in Russia appear less competitive in comparison to local dairy originating beef or imported Brazilian beef [45].

New EU dairy policies involved the suppression of milk quotas from April 2015. Consequently, raw milk prices dropped dramatically from a peak at the end of 2013 at EUR 0.40 per kg to a historic low in 2016 at around EUR 0.25 per kg. The Russian import ban on EU dairy products from August 2014 further compounded the low milk prices in the EU market [46]. The milk price crisis over that period induced increased slaughter and growth in the proportion of dairy cows in the meat supply chain. This lowered beef producer prices, with the Europe FPPI in 2016 24% lower than the 2012 peak, back to 2007 price levels. However, the pressure on beef producer prices was less severe compared to that for milk and recovered quickly in 2018 to be 16% below the 2012 record.

#### 3.1.4. The China FPPI

The China FPPI represents producer prices in USD, which are significantly correlated to that of the local currency (ρ value ˂ 0.001). As shown in Figure 6, the index increased until 2014 and has been less volatile (around a rapidly rising trend) compared to the other indices. Finished cattle prices in China have been mainly driven by the rapid growth in domestic demand for beef and rising cost of local beef production. Thus, they are less exposed to the global beef prices’ volatility.

After largely matching the upward trends in the Global FPPI in the 2000 to 2011 period, driven by domestic demand and supply changes, the China FPPI rose more rapidly than global prices between 2012 and 2014, a result of even faster demand growth and slowing production growth. However, since 2014, China prices have stabilized and largely followed the global FPPI, reflecting recovery in production growth and the rapid expansion in imports—which has both arrested the demand and supply imbalance and made China more sensitive to global beef trade prices. The price rise in 2019 was largely due to the increased national beef consumption, up 9% year-on-year, as beef was used to help fill the protein shortfall caused by African swine fever in China, particularly since summer 2019. Pork production in China dropped by 21% in 2019 compared to 2018 [21].

The growth in beef consumption in China has been mainly driven by the rapid development of the Chinese economy together with urbanization. Consequently, increased Chinese are entering the new middle class and increasing the proportion of meat in their diet [4,47].

Per capita consumption of beef in China showed a continuous increase in the last two decades—expanding by 47% from an estimated 3.93 kg per capita in 2000 to reach 5.8 kg per capita by 2019 [21]. The per capita consumption in China is still lower than that in many developed economies. Given the population growth in China, the continuous growth in per capita consumption is rapidly increasing the national beef consumption, making China the third largest world beef consumer after Brazil and the US. Total domestic beef consumption increased by 58% between 2000 and 2019. However, national beef production increased by only 26% during the same period [21]. The slowing of growth in beef production compared to consumption has led to a sharp rise in beef prices in China over the index period, as indicated in Figure 6 (despite the controlled opening to beef imports).

Chinese beef producers are facing many challenges, such as (i) a highly fragmented industry and traditional practices making the business less attractive to investors, (ii) relatively high input costs, mainly feed and land costs, (iii) epidemic diseases and natural disasters in Southeastern pastoral areas, and (iv) resource (particularly land and feed) and environment constraints [47,48].

Despite the recent emergence of larger cattle farming and processing companies in the major beef production regions, beef production in China is still dominated by the small size farming operations. These farms produce up to nine finished cattle for slaughter per year and represent almost 50% of the beef cattle production in China. Furthermore, cattle finishing is concentrated in the cropping regions, the central plains and the northwest; thus, feed inputs are based mainly on home-grown grains and crop residues. Producers in pastoral regions in the northwest, where cattle herds are distributed over large areas, graze cattle on pastures with limited supplementation [47,48].

The growing demand for beef in combination with favorable producer price levels have resulted in increased cattle slaughter. The percentage of cattle slaughtered relative to the total cattle inventory increased sharply during the index period, reaching nearly half of the total cattle in 2019. China’s total cattle herd diminished by 26% in the last two decades, while the national suckler cows inventory dropped by 28% over the same period [21]. The shrinkage in the national cattle herd has reduced the production capacity of the Chinese cattle industry. This in turn induced a continuous rise in producer prices and the opening to beef imports—transforming the country into the world’s largest beef importer with 1.6 million tonnes of imported beef in 2019, up from 0.35 million tonnes in 2014. Brazil, Uruguay, Argentina, and Australia are the top four suppliers to China, followed by New Zealand and Canada [48].

### 3.2. The Global Weaner Cattle Producer Price Index WPPI

The global WPPI measures weaner calf prices as the final output sold by cow-calf farms to pre-finishing operations (backgrounding or stocker) or directly to cattle finishing operations. In beef fattening operations, beef production starts with the purchase of surplus dairy animals for fattening or weaners from cow-calf farms that are mostly pastoral-based [30,31].

The WPPI is of particular importance to cow-calf farmers (output prices) but also to finished cattle producers (input prices). Thus, the index is an intermediate input price index in terms of the entire cattle finishing business. It is known that weighted averages of the differences between output and intermediate input price indices ideally reflect the value-added producer price index [13]. Of interest to producers is deflating changes in value added over time. Therefore, bringing the FPPI and WPPI together, as shown in Figure 7, may ideally serve to monitor changes in value added for finished cattle producers over time. This is of major importance to feedlot and pasture production systems in which weaner purchase costs constitute the largest proportion of total finished cattle production costs [49].

As shown in Figure 7, the WPPI and FPPI follow each other closely, as expected given that the WPPI is an input cost of the FPPI. Hence, the factors driving the initial rise in the WPPI to a peak in 2014 and the fall since are essentially the same as those outlined for the FPPI previously in Section 3.1; thus, they are not repeated here. However, significant (partly cyclical) shifts in these price relativities can be expected due principally to biological production lags, seasonal conditions and movement in feed costs. Changes in finished cattle prices, costs, and profitability should have an amplified impact on the price that cattle finishers can pay for weaner cattle, as occurs noticeably in the US (Figure 8). Furthermore, droughts impact cow calf operations, being pasture based, more than finishing, which causes larger shifts in weaner turnoff and, hence, weaner prices.

However, differences in the year-to-year movement of the FPPI and WPPI are only small. The principal example of this is the faster rise in the WPPI than the FPPI in the 2010 to 2014 period, of 44% and 27%, respectively. This can be attributed, to some extent, to the normal biological production lags, which delayed cow-calf producers’ response to the strong demand (and weaner prices) from a highly profitable cattle finishing sector at that time. In the US, for example, the herd cycle ended in 2014 with low beef production and record high beef (and finished cattle) prices. With the smallest cattle herd in half a century, and thus fewer weaner calves, this created extra competition between stocker, backgrounding operations, and feedlots for available feeder cattle [31]. US weaner calf prices, accordingly, rose faster than finished cattle prices between 2013 and 2015 (Figure 8a).

In addition to the gestation period, other constraints may have hampered the response of cow-calf production to high demand for weaners in finished cattle operations. The low cost for pasture and harvested forages in comparison to grains at that time may have provided an incentive for cow-calf producers to grow cattle for longer periods before placing them into feedlots—further reducing weaner sales and raising weaner prices [28,31]. This is an example of how climate and pasture conditions and relationships between the price of grain and forages can shorten or lengthen the age of entry into feedlots, and impact the time needed for raising calves to market weight. In contrast to the above example, the absence of adequate rainfall reducing the quantity and quality of forages produced can dictate premature termination of the grazing season and force cow-calf producers to market cattle earlier into feedlots to be fed—increasing weaner sales and lowering prices. Similarly, when forage prices are high relative to grain prices, a higher proportion of feeders enter feedlots directly.

Besides the impact on forage availability, drought has a direct impact on the herd fertility particularly in pastoral-based production systems. Brazilian pasturelands went through a widespread drought in 2015 during the mating season, impacting herd fertility [50]. This negatively impacted the calf crop and thus kept the weaner price index at high levels in 2015, whereas the finished cattle index decreased more over the period due to increased cull animal slaughter (Figure 8b).

### 3.3. The Global Lambs Producer Price Index LPPI

The Global LPPI includes the weighted price of slaughter lambs from 11 countries. The index has been driven mainly by slaughter lamb prices in Oceania, representing 56% production weighting on average over the base period, followed by Europe and Iran, representing 31% and 11% of global production shares, respectively (Table 2).

As presented in Figure 9, the LPPI has increased 232% from 2000 to its first peak in 2011—driven by a strong rise in global demand and constrained supplies from key producer countries. It then lost about one-third of this price gain by 2016, before recovering to again approach its 2011 peak in 2019.

The price of slaughter lambs in Australia and New Zealand has the main influence on the Global LPPI, reflecting their combined 56% production weighting. This share has been largely unchanged since 2000, though within this Australia’s share has risen from 25% to 32% over the index period while New Zealand’s share has fallen from 28% to 23%.

The Australian sheep sector is characterized by a steady increase in lamb production at the expense of mutton due to the shift from wool to meat breeds, which give better returns on the world market [51]. The developments in local lamb prices in Australia and New Zealand were highly correlated with each other and to their respective USD prices (ρ value ˂ 0.001).

Figure 9 shows both the similarity in movements of the Global LPPI and prices in Oceania, but also the slightly more volatile prices in Oceania, being strongly export focused, especially since China emerged as a major lamb importer in 2010. Besides, drought waves have exacerbated an already volatile sheep sector (in terms of production and prices) in Australia and New Zealand over the last two decades.

The fall in the LPPI from 2011 to 2016 followed Oceania supply growth and a temporary easing in Chinese demand growth. The recovery in the index in the three years to 2019 reflected price rises in Australia, New Zealand, and Germany, more than offsetting price falls elsewhere in Europe. The price rise in Oceania was assisted by stronger Chinese demand and import growth and lower supplies in Australia (due to drought).

Following a series of dry seasons in Australia and three years of economic instability on the back of the global financial crisis, good drought-breaking rains in 2010 and 2011 gave Australian sheep farmers the chance to rebuild their flocks [52]. This initially pushed up slaughter lamb prices to unprecedented levels, as more ewe lambs were held back for rebuilding and those sold were in strong demand from restockers.

Drought is the factor having the most pressing short- to medium-term impact on the Australian livestock industry over the past 20 years—as three of the seven severe and prolonged national droughts since 1885 have occurred in those two decades (the index period). These were the 9-year Millennium Drought of 2001–2009, the 4-year drought of 2012–2015 and the latest 2-year drought in 2018 and 2019, all affecting the key sheep regions. These three droughts contributed to falls in the flock of 39%, 8%, and 11%, respectively, as the consequent water and feed shortages affected flock productivity and forced Australian lamb and sheep farmers to de-stock sheep. Such a substantial fall in stock numbers led to lower sheep and lamb turnoff, due to lower numbers of lamb born, which was made worse in the few non-drought years by intense producer demand for available breeding stock to rebuild. This was particularly acute in 2017 and 2020 following the breaking of the 2012–2015 and 2018–2019 droughts—as many sheep producers moved to rebuild their depleted flocks encouraged by a good season, lower grain prices, and big harvests. This resulted in a fall in lamb supply and record prices in 2017. Australian lamb producer prices rose 17% in 2017 compared to 2015 (in USD), to be 98% higher than the average in the 2000–2009 period, but still slightly lower than the previous record in 2011.

Drought began to re-emerge late in 2017 and grew in coverage and severity through 2018 and into 2019 to become one of the most widespread and severe 2-year droughts in history. Lamb slaughter in 2018 was 8% above 2017 levels as producers again began liquidating flocks [21]. Nevertheless, lamb producer prices continued to climb by around 22% between 2016 and 2018, mainly due to record imports of both Australian lamb and mutton by China.

The Australian red meat industry is heavily reliant on international trade. The high proportion of production exported indicates that lamb and sheep prices are sensitive to any changes in global trade flows, particularly to China, the Middle East, and North Africa. Export demand for Australian lamb has strengthened remarkably over the last decade, driven by import demand from the Middle East and Asia, mainly China—commencing in 2010 but particularly since 2013 [53,54].

New Zealand significantly contributed to the lamb price peak in 2011. The New Zealand lamb and sheep flock declined by 20% from 2007 to 2011, with more intense competition for limited land due to expansion of the dairy sector. The New Zealand flock has continued falling but less severely, by 10% between 2011 and 2017, helping to keep local producer prices at higher levels than before 2011 [55]. Moreover, New Zealand lamb production is predominantly pasture based which makes the lamb crop highly impacted by droughts. For instance, 2013 recorded a sharp decline in the lamb crop in New Zealand due to lack of rain, especially in the North Island, which diminished feed supply and reduced stock condition at mating [56]. New Zealand’s exports grew significantly from 2011 led by demand growth from China, which overtook the European Union (EU) as NZ’s top market. The domestic consumption share of production has declined from 21% in 2000 to 5% in 2017 [55]. This makes the production prices highly vulnerable to the vagaries of the global market.

The Europe LPPI was moving with less volatility over the index period than the Oceania LPPI (Figure 10), indicating much more stable lamb prices in Europe than in Australia and New Zealand. This reflects the reliance of Oceania on the unstable global trade relative to Europe’s focus on more stable domestic demand and government supports, protected from volatile international trade prices by high import barriers.

Sheep production systems in Europe tend to be small, high-cost, and diverse compared to those in Oceania, resulting in low profitability, despite government protection and payments. Consequently, there has been an average 1.9% per annum decline in the EU production over the last two decades. Furthermore, lamb has increasingly become a marginal meat in the consumer basket in Europe, seasonally consumed at Easter and Christmas [51]. Per capita consumption in certain major meat markets, such as Germany and Spain, only averages 1 kg per annum, and in France it was down to 2.5 kg in 2019 from 4.6 kg in 2000 [21].

The United Kingdom of Great Britain and Northern Ireland (UK) is the only bright exception in the region, with its well-developed lamb and sheep production. The UK still retains the largest sheep population and sheep meat production in Europe (Table 2). UK lamb exports have been mainly consigned to other EU markets [51]. This brought greater stability to lamb producer prices in the Kingdom and, thus, to the Europe LPPI. Recently, New Zealand has reduced its traditional exports of frozen meat to the EU and increased exports of fresh or chilled meat. This significantly impacted the EU fresh-meat market and put more pressure on prices paid to European lamb producers [57]. This helps to explain the fall in the Europe LPPI below 100 index points since 2015.

The Global LPPI continued its upward trend in 2019, despite the stability of the Oceania and European LPPIs, illustrating that major changes in other countries can have a significant impact on this index. In this instance, it was the sharp jump in the Iran LPPI which increased by 75% between 2017 and 2019 (123% for local prices). The Iran LPPI showed a constant and exponential increase since 2014 (Figure 11). This was mainly attributed to political and macro-economic conditions resulting in massive inflation and currency devaluation—the Iranian rial lost nearly four-fold of its value relative to the USD [36]. This in turn pushed up production costs of sheep farming, particularly feed costs, because imported feed became expensive. Nomads keep 59% of the sheep population in Iran on extensive free communal grazing lands [58]. Widespread droughts in the past two decades have diminished pastures and grain harvests, requiring a greater reliance on imported feed [59,60]. Exchange rate developments made it more difficult for lamb and sheep producers to get imported feed on time and at a reasonable price. These issues have been making livestock production less profitable than ever, pushing producers out of business, particularly in pasture-based production systems. In the past two decades, the country’s sheep flock suffered a significant decline, falling 14% from 51.7 million in 2000 to 44.7 million in 2019. Lamb production shrank by 20% over the same period [21].

### 3.4. The Global Sheep Meat Producer Price Index SPPI

In many countries, there is no data distinguishing lambs from sheep meat, instead using the category “sheep meat produced” or “mutton” referring to lambs, culled ewes, and wethers. In China for instance, given the importance of wool production, mutton is the main meat output versus lamb [55]. Hence, whereas the global LPPI has 11 contributing countries, the Global SPPI includes weighted prices from 20 countries (Appendix A, Table A1). As shown in Table 3, the Global SPPI is dominated by sheep meat prices in China, contributing 60% of the global sheep production share, followed by Australia (10%) and New Zealand (6%).

China’s production share among the index countries has risen from 48% in 2000 to 62% in 2019, with commensurate falls in the shares of all other major producers. In contrast to the Global LPPI, where China is not represented, the Global SPPI kept rising beyond 2011 to peak in 2014, at an index value of 112, followed by a two-year decline to 90 in 2016, before again exceeding the 2014–2016 base level (100) in 2019 (Figure 12).

Although China’s sheep meat production rose by 60% over the 20-year index period, demand growth has accelerated more sharply, causing a rapid rise in China’s sheep meat prices to be among the highest in the world. Supported by growing production and imports, per capita consumption of sheep meat almost doubled over the same period.

Rising incomes and urbanization of China, the positive Chinese consumer attitudes toward sheep meat and the increase in outdoor meals has led to a surge in sheep meat demand. Away-from-home sheep meat consumption is estimated to account for 65% of the national total sheep meat consumed. This has mainly been driven by the widespread grilling of lamb leg and kebabs in summer, especially in northern China. Furthermore, high value cuts of sheep meat became common in high-end restaurants in larger cities. The growing demand attracted more farmers into the business, helped by low entry costs compared to cattle, while profitability has been high, given the elevated prices [61,62].

As demand for sheep products in China surged, new governmental policies were introduced in 2010, including grazing bans to prevent overgrazing in the key production regions of the northeast and western prairies. This caused strong competition for pastures and short feed supply. Production costs have increased sharply, driven by rising feed costs as well as fuel and labor due to growing incomes in urban areas. Moreover, the outbreak of ovine rinderpest has placed restrictions on the movement of live animals [51,61]. These circumstances brought upward pressure on sheep meat prices in China to reach a peak in 2014, up by 164% compared to 2010 levels (139% for local prices).

Producer prices in China declined in 2015 and 2016 as domestic supplies increased as a result of improved breeding performance and gains in feeding efficiency. The national and regional programs supporting large scale feedlot production, funding to improve pastures, and a breeding improvement program resulted in a quick rebound of the national flock size [51,61]. Compared to 2014, the national sheep inventory increased 7% and 4% in 2015 and 2016, respectively. Consequently, national sheep meat production increased by 13% over the two years to 2016 [21].

China’s rapid emergence to become the world’s largest sheep meat importer by 2012 (and further 144% rise in imports since 2012), has been a major factor in the surge in global sheep meat producer prices. China has caused products to be diverted from other markets and contributed to the lift in global sheep meat prices in all countries. The African swine fever and the consequent shortage of protein increased sheep meat prices further in 2019 not only in China but in all key producer countries.

Since China’s sheep meat prices dominates the Global SPPI and its movement, it is useful to also introduce a Global SPPI without China (Figure 13)—helping to both highlight China’s influence and build up a more differentiated picture on how sheep meat producer prices are developing outside China.

The global SPPI without China is highly influenced by Oceania, with a 40% the production weighting, followed by Europe, Algeria, and Iran representing 20%, 10%, and 9% shares of production, respectively. By excluding China, the index displayed the same movements as the Global LPPI, rising to a peak in 2011 before falling back to 2015 and recovering almost fully over the last four years. The almost identical movement in these two indices is testament both to the close relationship between the price of sheep and lambs and the dominance of Oceania in both indices without China.

The Global LPPI rose by 28% between 2016 and 2019, while the Global SPPI without China increased by 24%. The difference is attributed to the sharp increase of lamb prices in Iran over the period contributed much more to the LPPI than the SPPI (having a higher weighting in the LPPI), which was only partly offset in the SPPI by the 8% decline in sheep meat prices in Algeria over the same period (Figure 14).

### 3.5. Relativity of the Global PPIs to the FAO Export Meat Price Indices

The agri benchmark global PPIs are unique in representing prices of livestock as sold by farmers. However, FAO produces a well-known set of global food price indices, including for bovine meat (beef) and ovine meat (sheep meat). The FAO meat price indices are measures of the international meat prices weighted with the average export value shares for 2014–2016. Thus, they are exported reference prices rather than the price being received by producers. However, they have often been used to approximate movements in producer returns in the absence of global producer prices.

The FAO Bovine Meat Price index is calculated monthly based on the export prices of US frozen beef, Brazilian frozen beef, and Australia frozen 90% chemical lean (CL) manufacturing beef to the US. In July 2020, fresh and chilled products were added to the US and Brazilian prices [17,18]. Producer livestock prices and traded meat prices should move together and, hence, it can be instructive to compare movements in the two series.

As Figure 15 shows, the Global FPPI and the FAO Bovine Meat Price index follow each other closely, with a correlation of 96% on a calendar year basis since 2000. The Global FPPI was on a slightly higher level compared to the FAO Bovine Meat Price index from 2005 to 2014, probably due to the negative impact of BSE-related import bans on US beef export prices (and the FAO index) and the inclusion of China in the FPPI—as Chinese cattle prices rose rapidly over that period.

Figure 16 shows the agri benchmark global LPPI and the FAO Ovine Meat Price index. The FAO Ovine index is represented by prices of New Zealand export lamb of 17.5 kg carcass weight. Prices of Australia lamb 18–20 kg carcass weight were also added to the FAO index since July 2020 [17,18].

The LPPI and the FAO Ovine index move closely together and the FAO series is only slightly more volatile. The FAO Index was expected to be more volatile relative to the LPPI, given the relatively thin world lamb market (only 9% of global lamb production is traded) and the stability of lamb producer prices in Europe, which brings more stability to the global LPPI (given its 31% production share).

Figure 17 shows that the global SPPI and the FAO Ovine index also broadly move together but not as tightly as with the LPPI. However, the global SPPI moved higher relative to the FAO Ovine index and the global LPPI between 2011 and 2015, to a peak in 2014 (three years later than the FAO index and LPPI). This is due to the dominant influence of China on the SPPI, as China’s sheep prices rose rapidly over that period. This is illustrated by the inclusion of the SPPI without China on Figure 17, which moves more closely with the FAO Ovine index, sharing the same peaks and troughs.

## 4. Conclusions

The global producer price indices, as introduced by the agri benchmark Beef and Sheep Network, offer advantages for in-depth price monitoring and analysis and provide a basis for forecasting. Much can be gleamed from comparing global producer livestock prices with meat prices between livestock categories and both with and between regions and countries. Smaller countries can benefit from comparing their producer prices to the global indices and those of neighbors and competitors.

The close cooperation between the agri benchmark Beef and Sheep Network partners ensures the reliability of data sets for defining the per unit prices received by cattle, lambs, and sheep meat producers as farm gate prices. This needs an interactive networking to define the most prevailing product types for cattle and sheep and the key production regions in the network countries.

The global, regional, and national producer price indices provide researchers and interested parties with a detailed breakdown of price developments that allows them to better understand recent market developments and challenges and the competitive situation. Benchmarking a country and region’s price developments against other countries or regions and the global indices is essential to judge trends in local prices being received by producers—as an indicator of profitability and as a precursor to possible supply shifts. Thus, the indices reflect whether producers globally are receiving more, or less, for the commodity (cattle, lambs, sheep) and might increase or reduce production and investment accordingly.

This would be of more use if there were producer price indices for those enterprises that compete with cattle and sheep for land, especially dairy and cropping, and those that compete for the consumer dollar, such as pigs, poultry, and fish.

As feed is the second biggest cost in cattle finishing (behind livestock cost) in most production systems, producer indices for global feed prices can provide better understanding of the economic features of, and developments in, different cattle and sheep production systems globally.

The global farmgate cattle, lambs, and sheep prices have been mainly impacted by global supply and demand. Hence, prices in all major producing countries and regions tend to broadly follow global weighted average prices. However, factors which are more local in nature, such as natural conditions, exchange rates, policy interventions, and political and economic developments, can also have considerable impact on global and local price developments.

The study highlights the growing importance of China as a global beef and sheep meat producer and importer. This growth has been mainly driven by resource-related constraints limiting production capacity and the population becoming affluent with shifting to protein-based diets derived from animal products. The global beef and sheep meat trade flows have become increasingly targeted to a single country—China. This, in turn, points to the importance of a global set of producer prices indices in which China’s production and prices are represented.

## Figures and Tables

**Figure 1 animals-11-02314-f001:**
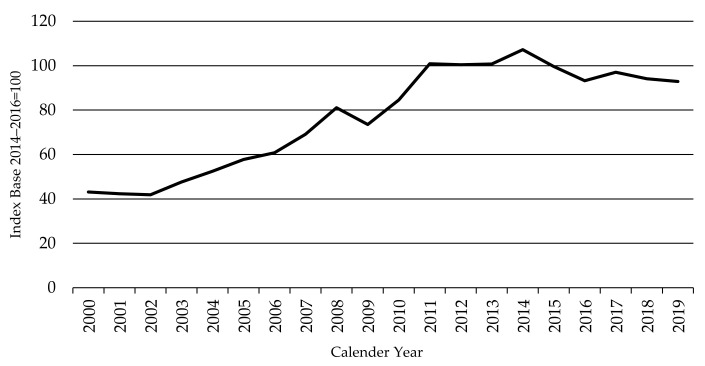
The Global Finished Cattle Producer Price Index FPPI, source: own illustration.

**Figure 2 animals-11-02314-f002:**
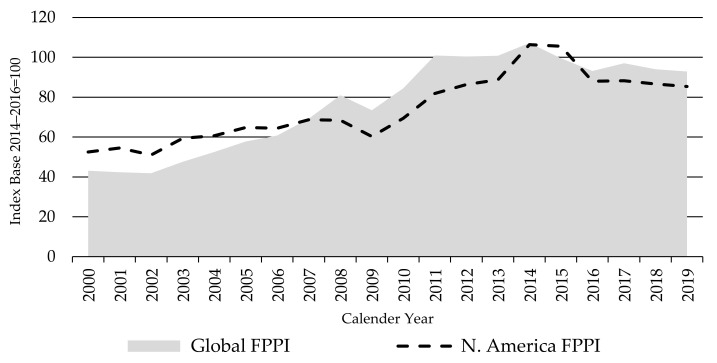
The Global and North America Finished Cattle Producer Price Indices FPPIs, source: own illustration.

**Figure 3 animals-11-02314-f003:**
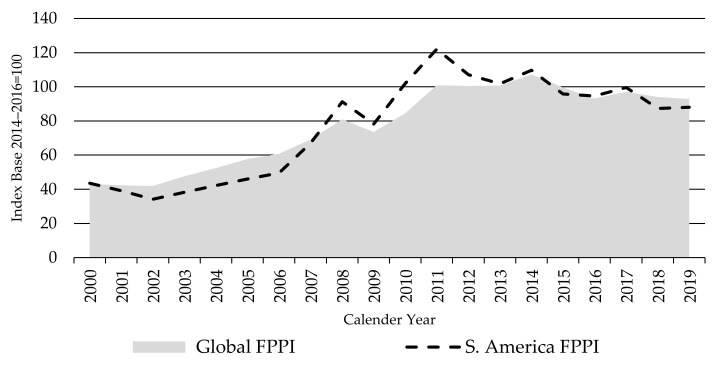
The Global and South America Finished Cattle Producer Price Indices FPPIs, source: own illustration.

**Figure 4 animals-11-02314-f004:**
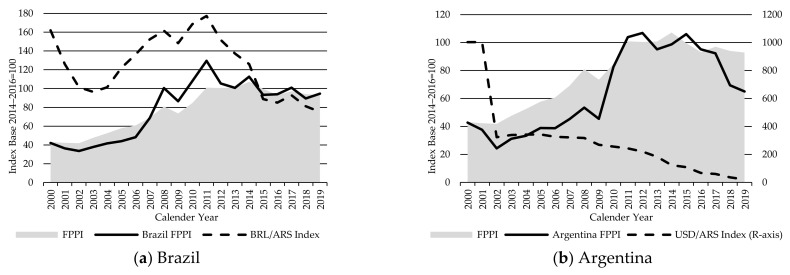
The Global Finished Cattle Producer Price Index FPPI vs. Brazil (**a**) and Argentina (**b**) Finished Cattle Producer Price Indices FPPIs and currency exchange indices, source: own illustration.

**Figure 5 animals-11-02314-f005:**
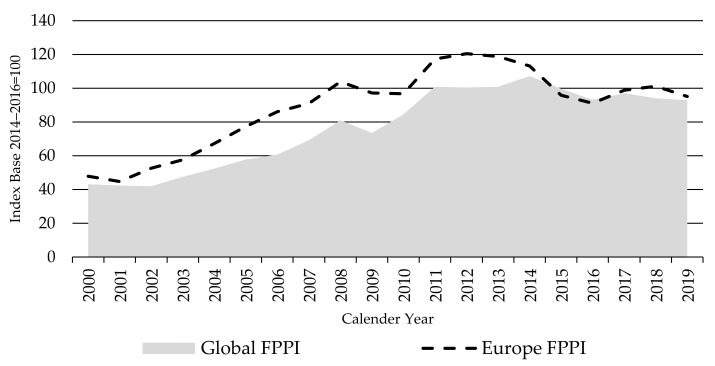
The Global and Europe Finished Cattle Producer Price Indices FPPIs, source: own illustration.

**Figure 6 animals-11-02314-f006:**
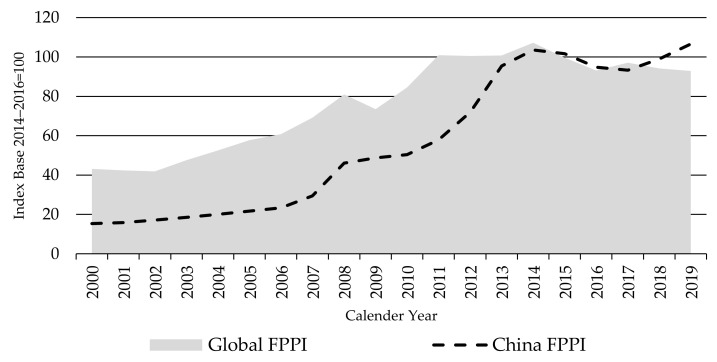
The Global and China Finished Cattle Producer Price Indices FPPIs, source: own illustration.

**Figure 7 animals-11-02314-f007:**
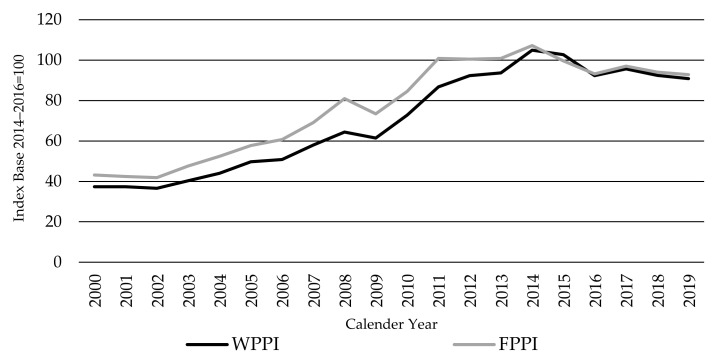
The Global Weaner Cattle Producer Price Index WPPI and Finished Cattle Producer Price Index FPPI, source: own illustration.

**Figure 8 animals-11-02314-f008:**
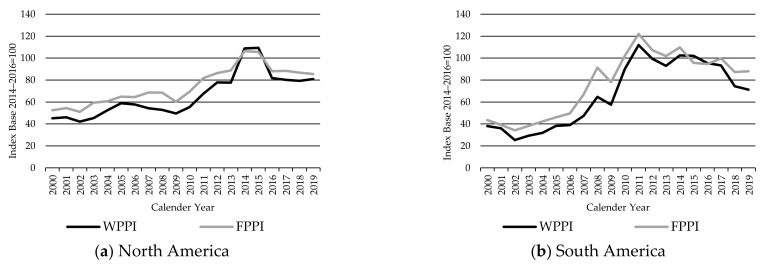
The Finished Cattle Producer Price Index FPPI and Weaner Cattle Producer Price Index WPPI in North (**a**) and South America (**b**), source: own illustration.

**Figure 9 animals-11-02314-f009:**
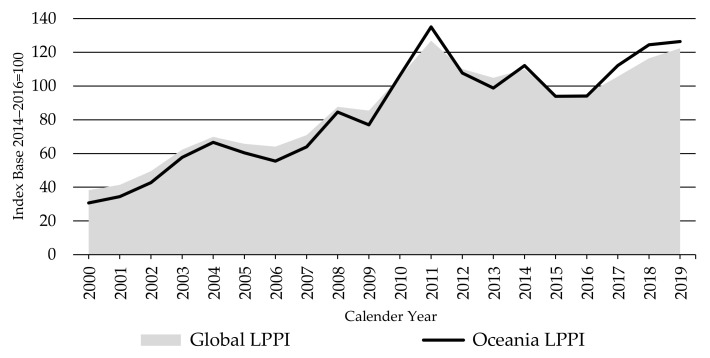
The Global and Oceania Lambs Producer Price Indices LPPIs, source: own illustration.

**Figure 10 animals-11-02314-f010:**
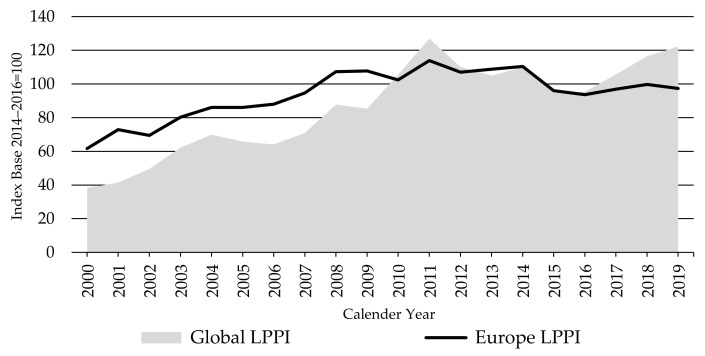
The Global and Europe Lambs Producer Price Indices LPPIs, Source: own illustration.

**Figure 11 animals-11-02314-f011:**
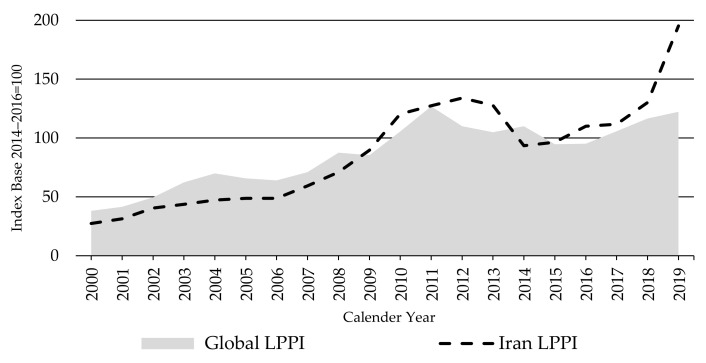
The Global and Iran Lambs Producer Price Indices LPPI, source: own illustration.

**Figure 12 animals-11-02314-f012:**
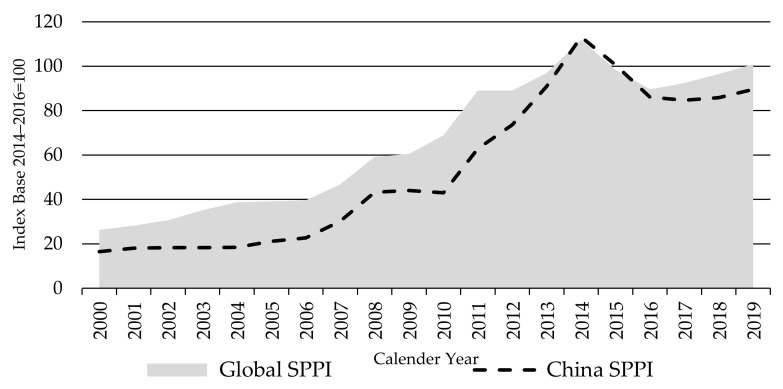
The Global and China Sheep Meat Producer Price Indices SPPIs, source: own illustration.

**Figure 13 animals-11-02314-f013:**
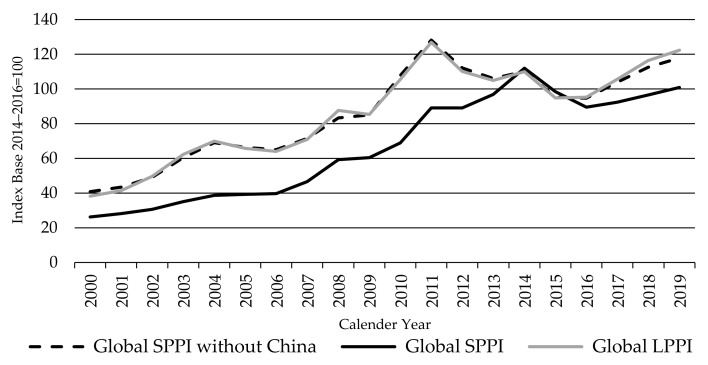
The Global Sheep Meat Producer Price Index SPPI without China, in comparison with the Global Sheep Meat Producer Price Index SPPI and the Global Lambs Producer Price Index LPPI, source: own illustration.

**Figure 14 animals-11-02314-f014:**
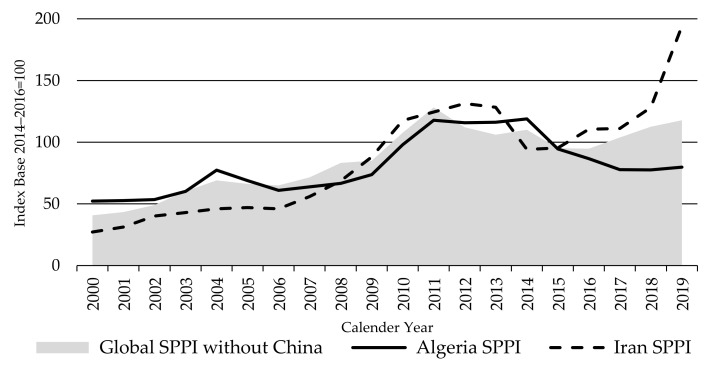
The Global Sheep Meat Producer Price Index SPPI (without China) in comparison with Algeria and Iran Sheep Meat Producer Price Indices SPPIs, source: own illustration.

**Figure 15 animals-11-02314-f015:**
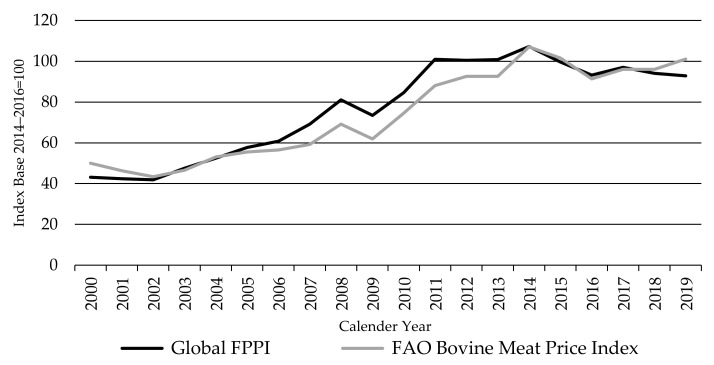
The Global Finished Cattle Producer Price Index FPPI and the FAO Bovine Meat Price Index, source: own illustration, FAO index [63].

**Figure 16 animals-11-02314-f016:**
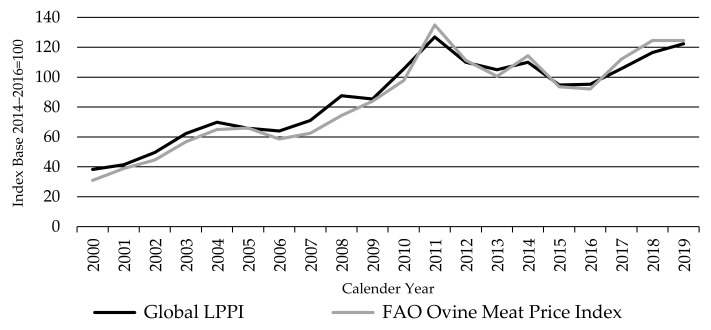
The Global Lambs Producer Price Index LPPI and the FAO Ovine Meat Price Index, source: own illustration, FAO index [63].

**Figure 17 animals-11-02314-f017:**
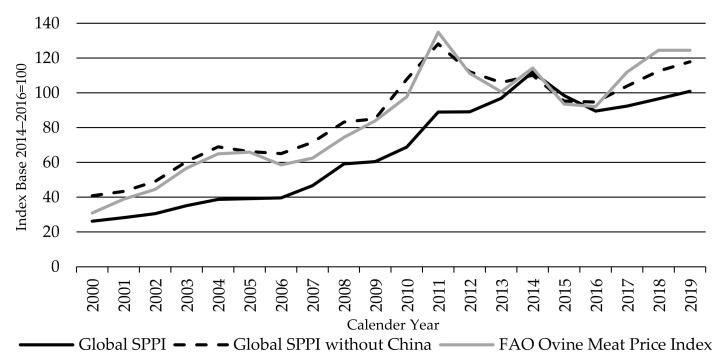
The Global Sheep Meat Producer Price Index SPPI, the Global Sheep Meat Producer Price Index SPPI (without China), and the FAO Ovine Meat Price index, source: own illustration, FAO index [63].

**Table 1 animals-11-02314-t001:** Overview of the introduced regional FPPIs, source: own illustration.

	**North America**	**South America**	**Europe**	**China**
Countries	CA, MX, USA	AR, BR, CO, PE, PY, UY	AT, CH, CZ, DE, ES, FI, FR, IE, IT, PT, PL, RU, UK	China mainland
Production share of total FPPI countries over 2014–2016	28% USA: 22%	28% BR: 18%	20% RU: 9%	12%
Prices’ correlation range (r) ρ value	0.74–0.97 ρ value ˂ 0.001	0.74–0.97 ρ value ˂ 0.001	0.81–0.99 ρ value ˂ 0.001	

**Table 2 animals-11-02314-t002:** Overview of the introduced regional LPPIs, source: own illustration.

	**Oceania**	**Europe**	**Iran**
Countries	AU, NZ	DE, ES, FR, IE, PT, UK	Iran
Production share of total LPPI countries over 2014–2016	56% AU: 32%, NZ: 24%	31% UK: 16%	11%
Prices’ correlation range (r) ρ Value	0.95 ρ value ˂ 0.001	0.80–0.96 ρ value ˂ 0.001	

**Table 3 animals-11-02314-t003:** Overview of the introduced regional SPPIs, source: own illustration.

	**China**	**Oceania**	**Europe**
Countries	China mainland	AU, NZ	DE, ES, FR, IE, PT, UK
Production share of total SPPI countries over 2014–2016	60%	16% AU: 10%	8% UK: 4%
Prices’ correlation range (r) ρ Value		0.95 ρ value ˂ 0.001	0.84–0.95 ρ value ˂ 0.001

## Data Availability

The data are not publicly available due to scientific cooperation agreements between the agri benchmark Beef and Sheep Network and the Institutional Partners.

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
