# Peer review of "Introducing the World’s First Global Producer Price Indices for Beef Cattle and Sheep"

_animals, 2021, doi:10.3390/ani11082314_

Round 1
Reviewer 1 Report
This is an interesting paper with an analysis for a producer price index that would enable producers to making management decisions be examining global prices. The authors provide significant context to understanding the fluctuations through historical narrative, localized cultural preferences, and supply and demand shocks of the time.
Below are a few points that need further clarification.
Would the global PPIs account for differences such as those arise from country specific product quality, industry concentration and structure, or asset fixity across different localities? If producers globally are to rely on this index for investment or divestitures, would issues such as those described above lead to over or under investment (or increases/decreases in herd sizes)?
While the 3 year base period is used to account for cyclical/seasonality - is this sufficient when the Authors mention that the US cycle lasted 10 years? Is this an issue in production conditions in other countries?
Is the data used in the calculation of the index proprietary? If so, how would private or public entities be able to calculate the metric?
line 847. Typo: "Much can be gleamed..." The author likely intended to use the word "gleaned."
Author Response
Thanks for your comments,
Please see the attachment

Reviewer 2 Report
Dear authors,
I have reviewed your manuscript and I would like to congratulate for doing this arduous work. The main strength of your manuscript is the novelty.
Your manuscript is well written. The methodogoly allows the repletion of the study and present new data that add new knowledge. In my opinion this manuscript can be accepted after minor corrections.
Line 41-46 Please include information about cattle distribution/number worldwide
Line 874 – Which factors?
Author Response
Thanks for your comments,
Please see the attachment.

Reviewer 3 Report
I added some comments in the authors' manuscript that I am attaching. A key point is that they do not explain (or at least I failed to find it) whether the prices are measured at current (not inflation adjusted) or constant (inflation adjusted), and whether they are at market exchange rates or at PPP values. Until they explain well this point the manuscript should be retained
My additional comments are a)the value of the paper is the effort to construct the price indices; the question is who/how will those indices be maintained and published; they should be open source; the data should be available.
b) The comments regarding that the movements at the level of producer prices are due to variables/causes X or Y, which are commented extensively in the paper, seem to be the result to qualitative analyses; although the comments on causality seem appropriate, they do not seem to be based on detailed quantitative/econometric analyses; there are some bivariate correlations with P values, but this is not a full econometric analysis. My suggestion is that in another paper they try a more formal econometric analysis.
c) the impact of exchange rates on domestic consumer and producer prices in developing countries has been largely documented; they may use the section on exchange rates in my book https://www.ifpri.org/publication/macroeconomics-agriculture-and-food-security-guide-policy-analysis-developing-countries

Author Response

(The authors gave the same response as above.)
